# Say Yes to Education—Buffalo: A Human Capabilities Approach to College Access and Local Economic Development

Nathan J. Daun-Barnett 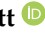

Graduate School of Education, State University of New York at Buffalo, Buffalo, NY 14260, USA; nbarnett@buffalo.edu

**Abstract:** In December 2012, researchers from the University at Buffalo partnered with Buffalo Public Schools and Say Yes to Education—Buffalo to assist students and families with the Free Application for Federal Student Aid (FAFSA). The community had just announced a last-dollar tuition guarantee for all public and charter high school graduates. Students had to apply for federal and state financial aid to be eligible. We use the human capabilities framework described by St. John to examine the contributions of this specific intervention and the broader collective impact strategy. In this study, we employ difference-in-difference regression analysis to examine the effects of a FAFSA completion intervention and find that providing support to students and families to complete the financial aid process increased FAFSA completion rates by more than 60%, year over year. In addition to considering the outcomes of this intervention, we report lessons learned in the process of establishing a university-community collaboration to improve postsecondary opportunity and economic development. We find that effective collaboration takes time and a shared commitment to understanding and addressing problems of practice in schools.

**Keywords:** university-community partnerships; college access; financial aid

## 1. Introduction

The city of Buffalo, like many postindustrial communities scattered across the Great Lakes and upper Midwest region of the United States (U.S.), suffered significant declines during what St. John (Ch. 1) describes as the global period, when neoliberalism replaced more progressive policies. At its height, Buffalo, NY, was a national center of commerce, serving as the gateway for the flow of goods from the interior portions of the country through to the Eastern Seaboard when the Erie Canal opened in 1825 [1]. Buffalo flourished through the middle of the 20th century and became a center for steel production. At the height of World War II, Bethlehem-Lackawanna Steel became the world's largest steelmaking operation, providing material supplies for the U.S. military effort [2]. As recently as the 1960s, the population of the city of Buffalo reached more than 580,000 residents [3]. Since that high water mark, the city declined in population to 278,379 [4] and grew increasingly segregated. Bethlehem Steel and the entire steel industry in Buffalo closed their doors along the shores of Lake Erie in the early 1980s. It would be fair to say that the city of Buffalo was in a period of social and economic decline for nearly a half-century. The past decade was very different for Buffalo and the Western New York region. The city experienced a renaissance in the early part of the 21st century. Part of that success is attributable to a comprehensive, community-based educational reform and economic development strategy driven by Say Yes to Education—Buffalo, a local non-governmental organization (NGO).

Say Yes Buffalo was launched in partnership between local civic, community, education, business, and philanthropic leaders and a national non-profit organization of the same name. The economic resurgence of the city and the region began before the organization started. Still, it was the driving force behind a comprehensive social change initiative to leverage the power of postsecondary education to drive economic growth in Buffalo. Say

Yes was instrumental in shepherding a large-scale social change initiative in Buffalo, and this paper focuses on the signature feature of their work—an endowed scholarship that was described as a last-dollar tuition guarantee. In December of 2011, when Say Yes to Education, the national non-profit organization, announced it was coming to Buffalo, they promised that every student who graduates from Buffalo Public Schools would be eligible for free tuition at any state 2-year or 4-year college or university or equivalent support at a network of private colleges. To qualify for this last-dollar tuition guarantee, students must complete the Free Application for Federal Student Aid (FAFSA), the New York State Tuition Assistance Program (TAP) application, and an online registration form for Say Yes. In this paper, I examine the effects of an intervention designed to help students and families complete the financial aid application process and effectively maintain their eligibility for the tuition guarantee. In the process, I address two key questions:

1.  Does the FAFSA completion project increase FAFSA completion rates for Buffalo public school students eligible for the Say Yes tuition guarantee?
2.  In what ways does a collaborative partnership among university researchers, public school leadership, and a local NGO enhance efforts to assist students with their financial aid application process?

The first question is the focus of the empirical analysis of this paper, while the second question provides an opportunity to think about the underlying collaboration that made the FAFSA completion project possible.

Buffalo is part of a region in the U.S. known as the rust belt, which was essential to the steel and automobile industries through the middle of the 20th century. It earned the name because it represented the literal and figurative decay of the community, as many steel-producing industries were exported to nations with lower production costs. In many ways, the new period of populist nationalism in the U.S. is in response to the failings of neoliberal policies during a period of globalization. Populism in the U.S. context was complicated by our struggles to reconcile our history of racism rooted in the slave trade during our nation's founding. Moreover, we witnessed evidence of this during the COVID-19 pandemic when decisions to ban travel from Muslims and East Asian countries were fueled, at least in part, by ethnocentrism and xenophobia. These debates continue as we struggle to redefine our immigration policies as migrants from Mexico and the nations in South and Central America seek refuge on our Southern border.

Buffalo is a symbol, in many ways, of the challenges and opportunities facing the U.S. in the post-neoliberal transition. The city suffered significant economic challenges due to globalization, the decline of the steel industry, and the subsequent white and middle-class flight from the city center to the first and second-ring suburbs, beginning during the civil rights era and the desegregation of schools. Today, Buffalo serves as a refugee resettlement community on the Northern border with Canada. In recent years, the city experienced modest population growth, primarily attributable to the influx of refugees from across the globe. Today, the city is home to a majority-minority population, where slightly less than 45% of the population identifies as White. The remaining population is comprised of Black and African American (33.3%), Hispanic or Latinx (12.2%), Asian (6.7%), and residents of two or more races (5.9%). Additionally, students in the Buffalo Public School (BPS) system speak 82 different languages, and more than 19% identify as English language learners [5], which indicates the role immigration played in the resurgence of the city of Buffalo.

In the next section, I describe the work of Say Yes to Education—Buffalo and describe their role using the Human Capabilities Framework as St. John described earlier in this volume. Say Yes is a community-based strategy modeled after several place-based promise programs designed to leverage the benefits of postsecondary education for local economic growth. Hundreds of communities across the country considered launching similar initiatives and dozens developed some variation of a place-based tuition guarantee program [6], only a few of which were as successful as Buffalo. I review relevant literature examining these community-based strategies to improve student outcomes in this section. The third section describes the FAFSA Completion Project and its role in providing a community-

based strategy to promote postsecondary opportunities. Many researchers noted that the rising cost of higher education in the U.S. became a significant barrier for low-income, first-generation, and racially minoritized students [7–12], and several identified the complexity of the financial aid application process as a contributing factor [13–21]. The research literature is mixed on the effectiveness of existing strategies to improve the financial aid application process. The intervention we examine responds to the work being carried out to simplify the financial aid application process in the U.S. and focuses more on helping families navigate the complexity rather than simplifying the process. The fourth section reports on the results of the intervention and the implications of the work in terms of the broader educational and economic development goals of Buffalo and the Western New York region. In the final section, we reflect on the implications of this work for local communities situated in other national contexts.

### 1.1. Say Yes to Education—Buffalo

Say Yes to Education is a national non-profit organization founded in 1987 by George Weiss, who built his wealth as a hedge fund manager [22]. Initially, Weiss emulated Eugene Lang's *I Have a Dream* initiative, promising a class of sixth-grade students in a Philadelphia elementary school a free college education. Weiss named the project in response to the famous "Say No to Drugs" campaign of the 1980s—his alternative for students was to "Say Yes to Education." Weiss brought his model to cohorts of students in other cities, including a partnership with the Harlem Children's Zone in New York. Say Yes differed from most philanthropist-driven tuition guarantee initiatives because it provided wrap-around support services to help students prepare themselves to take advantage of the promise.

In 2008, Say Yes launched its first citywide initiative in Syracuse, NY. Instead of sponsoring a cohort of sixth-grade students in a school or across a district, the new model was to build the capacity for the city to cover the cost of postsecondary education for every graduate of the Syracuse school district. This version of the Say Yes to Education strategy was announced only three years after the city of Kalamazoo, Michigan, launched the Kalamazoo Promise [23]. The shift from individual efforts of private philanthropy to the collective action of an entire city shifted the purpose from unique postsecondary opportunity to local and regional economic development. To secure the support of local business, civic, education, and philanthropic leaders, the strategy was linked to leveraging an educated workforce to catalyze economic growth for the city and the region. Kitchens [24] described how the Kalamazoo Promise was one of five distinct strategies designed to grow the local economy, recognizing that in a knowledge economy, a college-educated workforce is a necessary first step to growing the local economy.

Local leaders in Buffalo were following the evolution of the promise community models and considered the Kalamazoo approach but chose to partner with Say Yes to Education, mainly because the model included both the tuition guarantee and a comprehensive approach to supporting students and families throughout the journey from kindergarten through the attainment of a college degree. The Syracuse chapter of Say Yes to Education focused on reforming the local school district [25] and relied heavily on the Chancellor of Syracuse University to accomplish its postsecondary objectives. The university and community partnership became a cornerstone of the model and continued to be the case when Say Yes announced its partnership in Buffalo.

Say Yes to Education Buffalo is organized as a collective impact strategy for social change, where the focus is on developing a broad partnership of local leaders who can set a shared agenda, engage in mutually reinforcing activities, utilize data to examine the collective effects of the work, and support continuous communications among critical stakeholders. A central feature of successful collective impact strategies is the existence of a backbone organization whose sole responsibility is to manage the shared agenda for the partnership [26,27]. As Kania and Kramer (2013) noted, the backbone organization is critical because every other stakeholder has its agenda and cannot easily take on the priorities of the collective when the plans may not overlap.

The tuition guarantee is the most prominent feature of the Say Yes to Education collective impact strategy. However, as George Weiss and others realized, financial resources are necessary but insufficient to ensure students can take advantage of promise programs. To illustrate, in 1999, the Detroit Pistons won the National Basketball Association (NBA) championship, and in response, the Detroit News published and sold commemorative programs to the fans of Detroit. The Detroit News used the proceeds to sponsor a free college education for sixth-grade students in a local middle school. In 2005, the News ran a story following the classroom of students they sponsored six years earlier. Of the 20 students in the class, five could no longer be located, and only two were planning to attend college in the fall—in short, only 10% of students were prepared to take advantage of the promise with no other resources or support to help them along the way.

Say Yes to Education provides a generous tuition guarantee, but their support does not stop with the scholarship. They include access to health clinics, a legal clinic, access to mental health support, individual social workers placed in every school to address the emergent needs of students and their families, parent resource centers in schools staffed by college navigators, and robust campus-based support for students who make the transition to the local colleges which educate the most BPS graduates. Additionally, Say Yes provides leadership on an array of additional supports typically provided by the school district, including extended learning programs after school and on weekends, peer mentoring, and a program called Breaking Barriers, designed to support boys and men of color in the schools and the local community. One additional wraparound service provided as part of the Say Yes to Education model is the focus of this study—the FAFSA Completion Project [28].

The work of Say Yes, even from the earliest stages of its organization, reflected the tenets of the human capabilities framework (St. John, above). While Say Yes—Buffalo was not meeting families' complete subsistence needs with organizational resources, they partnered with schools and other community-based organizations to improve postsecondary opportunities for low-income families. Over time, the organization evolved to pay more specific attention to students' postsecondary pathways into college and careers, providing information to students and families about the process and creating sustainable networks to support postsecondary participation and success. The collective impact model that shaped their citywide initiatives is consistent with the community support networks designed to address various family needs, prepare students for college, and provide the resources the community needs to help students navigate the transitions into and through college.

### 1.2. The FAFSA Completion Project

The cost of college is a significant barrier to postsecondary access for many students and families—particularly those from families with lower incomes and limited intergenerational wealth. According to the most recent trends in college pricing reports, undergraduate tuition ranges from $3800 at community colleges and $10,470 for in-state public four-year colleges to more than $38,000 for a private non-profit college [29]. These prices do not account for the cost of living, which is approximately $13,000 per year for room and board. To put these numbers into perspective, the median household income in the U.S. in 2021 was $70,784 [30], and a low-income family would earn less than 200% of the federal threshold, the equivalent of $52,492 for a family of four in 2020. Sending a child away to the public, in-state, four-year college would cost nearly $25,000 per year, nearly half of what a low-income family would earn in a year. Fortunately, the federal—and many of the state—financial aid programs are means-tested, and so, lower-income families receive more significant financial assistance. The challenge is applying for the financial aid they need.

The Advisory Committee on Student Financial Assistance [31] identified the need to simplify the financial aid application process. It made ten recommendations to Congress, including creating a more straightforward form, eliminating unnecessary questions, reducing reliance on paper applications in favor of a completely online system, and allowing students to apply for aid sooner to have better information in the college decision process.

Many of those early recommendations were adopted, but the FAFSA continued to be a complex application for students and families. Dynarski and colleagues [10,18,19] continued to call for a more straightforward application, arguing that complexity in the financial aid system was a barrier to postsecondary participation and that the form could be reduced to as few as four or five questions that would provide reliable estimates of family need. Over the past decade, the Department of Education simplified the form and dealt with using tax information in two ways. First, they now allow "prior-prior" year taxes to be used in calculating a student's expected family contribution (EFC) so that the application can open sooner for students. Second, they built the Internal Revenue Service (IRS) tax retrieval tool to allow families to import their tax information into the form automatically. Creating a more straightforward form is an effective policy response but not a panacea. For example, the U.S. Department of Education added more questions to determine eligibility for independent student status, recognizing there are a variety of reasons why a student may not be supported by their parents to pay for college, including homelessness, participation in foster care, the death of their parents, or having and supporting their children. In those situations, increasing the number of questions helped to make the process more equitable. To this point, scholars discussed the importance of simplifying the financial aid application process, and they were effective advocating for change, but they did not conduct empirical research to examine the effects of those efforts to simplify the application.

Another line of inquiry focuses on providing direct support to students and families to complete the financial aid application process. This body of research recognizes that it may not be feasible to make the process simple enough, so they focus on strategies to help students and families complete the complicated processes. The College Goal Sunday initiative was created to assist families with the financial aid application process. The campaign name is a play on "Superbowl Sunday," which is the most popular sporting event in the U.S. that occurs each year in late January or early February—when most families would complete their financial aid forms (until the recent shifts to prior-prior year taxes and the October 1 opening of the form). In a typical event, financial aid counselors from local colleges would partner with school districts in their service area and identify a day (Saturday or Sunday most commonly) when families could sit and fill out their FAFSA and any relevant state forms with a financial aid expert. An evaluation of the program found that, in a typical year, these events served nearly 40,000 students but did not reach the target audience—low-income families who were least likely to apply for financial aid or attend college [20]. The program was effective for some, but it was not able to reach those with the greatest need. It was largely a descriptive study that suggested mixed effects overall but little success reaching the population of students who need financial aid most in order to attend college in the U.S.A.

Bettinger and colleagues [13] took a slightly different approach to FAFSA simplification, and their work was a precursor to the IRS tax retrieval tool. In 2008, they partnered with H&R Block tax accountants to conduct an experiment testing three conditions for FAFSA completion. Eligible families were sorted into three randomly assigned groups: (1) the control group that had their taxes completed, (2) the information-only group where families completed their taxes with an expert and were given information about how to file the FAFSA, and (3) the FAFSA completion group where the tax data were automatically transferred into the FAFSA online. The findings were striking. First, the information-only group was no different from the control group, meaning they were no more likely to apply for financial aid or attend college than those who received no information. This was a significant finding on its own because the most common FAFSA intervention in schools is a financial aid night where a local expert conducts a session for students and parents to educate them in the financial aid system and the application process—essentially, the information only treatment and the study found that it was not an effective strategy. The more important finding was that students in the whole treatment group were much more likely to apply for financial aid and to attend college. In some ways, this was a more robust version of the College Goal Sunday initiative because it had both the expertise of tax

accountants and the automatic import of tax information into the FAFSA. Both initiatives focused on simplifying the process by helping families navigate the complexity rather than attempting to simplify the form. The limitation of the H&R Block approach was that it is a fee-for-service model, and many low-income families cannot afford it or do not file taxes at all. They also mainly served independent students who did not need to use their parents' taxes to file their FAFSAs. Our intervention builds upon both the College Goal Sunday and the H&R Block interventions. We provide a comparable support to that of the College Goal Sunday, but we partnered with school counselors during the school day to reach students who would not otherwise seek out the support.

In Buffalo, we also built from the strengths of the Bettinger model but adapted it to target high school graduates financially dependent on parents or guardians while providing free tax support. Before developing the program, I spent six months shadowing school counselors at one of the lower-performing high schools in Buffalo. During that time, I was interested in understanding what barriers prevented low-income students and students of color in BPS from choosing to attend college. I found that counselors were asked for a considerable amount of help from their students on the administration of the college choice process—the college search, college applications, campus visits, SAT or ACT registrations, fee waivers for eligible students, and the federal and state financial aid forms. After that period of exploration, I proposed several strategies to free counselors from the administrative burden of the college choice process so they could spend more time counseling students on their future career paths.

In January 2010, we launched a simple pilot project at another comprehensive high school in Buffalo. We partnered with the United Way to make the voluntary income tax assistance (VITA) program accessible to students and families. At the same time, the university would provide trained volunteers to assist with the FAFSA and TAP applications. In just two months of service, our volunteers assisted 31 students with their financial aid applications—a modest number, but it was two-thirds of all the FAFSA applications submitted at that high school. We also found that the VITA program was not a successful addition to the project. Very few of the families came in to use the VITA service, and we did not have the technical capability to automatically import tax information into the FAFSA, as Bettinger was able to. Perhaps the most important finding from the school counselor's perspective was that it took approximately 90 min per student we served, which was effectively 45 h of their time that we could give back to them.

Less than a year later, Say Yes announced they would begin operating in Buffalo and offer a tuition guarantee to the next graduating class. District leadership asked their school counselors what could help more families complete the FAFSA and TAP applications. The counselors identified the pilot project as a possible solution. To bring the program to scale, we began by recognizing what was already being offered in the Buffalo schools. For several years, the district hosted a College Goal Sunday event which assisted approximately 110 students, on average, with the financial aid application process. The event accounted for over 20% of the FAFSAs submitted in Buffalo Schools in 2012.

In consultation with Say Yes and Buffalo Schools, we created a three-phase project to assist students and families with the financial aid application process. In the first phase, we went into each of the 14 participating schools, met with classes of students in a computer lab, and helped students complete the first half of the process, which, at that time, included generating their personal identification number (PIN) and completing the student information sections of the FAFSA, including their demographic information, high school attended, list of possible colleges, and responses to the independent status questions. The second phase was the College Goal Sunday event, which BPS called the Scholarship Fair. At this event, we integrated the tax prep services from VITA with financial aid support from local financial aid counselors. The final and longest phase of the project brought individual volunteers into each high school once or twice per week to work individually with students and their families on the remaining portions of the form. Say Yes announced a deadline of April 1 for financial aid forms to be submitted to be eligible for the tuition

guarantee, and so, the project was active from the middle of January through the end of March. From our experience shadowing school counselors in the year prior, we understood the importance of tailoring the project to each school; so, at the beginning of January, we met with each school counseling team to learn how best to implement the program in their schools, and we made slight modifications to the classroom/computer lab portion at the beginning and the final phase working with individual students. We understood that the program would only be effective if the counselors believed in it and were willing to connect their students to project volunteers.

## 2. Materials and Methods

In this study, we were interested in examining the effects of the FAFSA completion intervention on the submission of financial aid applications. Our primary challenge was isolating the FAFSA Completion Project's effects from the scholarship announcement. Both interventions were likely to impact financial aid application behavior, and it can be challenging to assess the contributions of each. As with most education research, we could not randomly assign students to treatment and control groups in this situation, and so, we relied on several comparisons to help establish the FAFSA project's effects. First, we compared FAFSA completion rates from Buffalo to those of comparable, mid-sized urban centers across New York State. The tuition guarantee applied primarily to New York State public colleges and universities, and so, this contrast helped to assess the extent to which financial aid application behavior changed in Buffalo compared to peer cities. It did not indicate the relative contribution of the tuition guarantee or the application support. However, it did illustrate that something different was happening in Buffalo, and it was not the result of changes happening across the state.

The second set of analyses compared the FAFSA completion behavior in BPS to their public charter school peers. The advantage of this comparison was that both BPS and charter school students were eligible for the tuition guarantee. However, in the first year of the project, the FAFSA Completion project only assisted students in BPS, meaning charter school students provided a useful counterfactual. In later years, we expanded the project to include charter schools. However, the first year allowed us to examine the extent to which the FAFSA project had an independent impact on financial aid application behaviors beyond the announcement of the tuition guarantee. It is important to note that the study reports the total number of applications completed per school rather than the percentage completed because it was difficult to identify the appropriate denominator. At the time of the intervention, all seniors and any juniors identified as eligible to graduate early could utilize the services of the FAFSA Completion project. However, not all eligible students graduated from high school, and not all graduates intended to go to college. So, the possible denominators would be the number of currently enrolled seniors, the number of high school graduates in the cohort, or the number of graduates who intend to enroll in college—a measure for which reliable data were unavailable.

### 2.1. Data Source

The year we began the project, the U.S. Department of Education published FAFSA completion numbers by high school and state bi-weekly. These data reports included the number of FAFSAs submitted and the number of FAFSAs completed (accepted by the federal government) during the current filing year by the date the data were made available. They provided comparable data for the same school and date from the prior year. The Education Department (ED) continued making these data available each year so that counselors, schools, and college-access programs can track their progress at the school level. The bi-weekly data remain useful to examine completion trends throughout the project and compare them to the same time points in the previous year for Buffalo Schools. For the regression analyses, the data from the April 12 data file were used because it was the final period available at the end of the project.

The data are publicly available through Federal Student Aid [32], and they were extracted from submitted FAFSA forms by linking the student to the high school from which they graduated, identifying the first time a student was ever enrolling in college, and limiting the age of students to no greater than 19. These data are appropriate for comparison purposes across schools in the aggregate, but they were likely to under-report total numbers for two reasons. First, at the time of this project, students could type in the name of their high school and submit without verifying the school from the list of identified schools in the application. Those applications cannot be assigned to the appropriate school or districts and do not appear in the data. Second, it is not uncommon for a student in BPS to graduate after age 19. Students older than 19 will not appear in the aggregated high school numbers. This is a limitation of the data, but we expect similar patterns across all high schools and districts included in the study because we are comparing schools and districts serving similar demographics of students. Each school/year was one case in this analysis, and there were a total of 44 high schools across the four urban centers, resulting in 88 school-year cases. Only two years of data were used for two reasons. First, we only had access to one year of data before the implementation of the project, and so, we wanted the sample to be balanced in terms of treatment and control group sizes. Second, it was the first year of the intervention, which was ideal for assessing the effects of the intervention, independent of the ways a system may adapt over time.

*2.2. Analytic Method*

We employed difference in differences (DID) analysis to consider whether the FAFSA completion intervention had an effect independent of other state-level factors or the announcement of the Say Yes to Education scholarship. The DID method is an econometric tool that allows researchers to more closely approximate experimental treatment and control groups when random assignment is impossible. A simple time series difference analysis considered the outcome before and after the policy implementation and assumed any measurable difference was a consequence of the policy. However, there may be something unique about the schools where the intervention is implemented. The difference in differences approach allows researchers to use non-treated groups as a counterfactual. The difference between the treatment and control groups before the treatment was removed and the observed difference post-treatment was attributed to the intervention. The DID model assumed that the trajectories of the treatment and control groups were parallel in the absence of the intervention and that differences between the two groups after the pre-treatment differences were removed attributable to the program. We were unable to test the parallel trends assumption, given our data constraints, and so, we added several controls that may account for sources of variation, including school size, percent of students eligible for free or reduced-price lunch, the percentage of students of color enrolled in the school, and the percentage of students suspended in the school in a year. Each school's controls were available through the New York State Report Card.

The difference in differences model takes the form:

$$y_{it} = \beta_0 + \beta_1 X_i + \beta_2 T_t + \beta_3 X_i \times T_t + \varepsilon_{it} \tag{1}$$

In Formula (1), X is a dummy variable for assignment to the treatment group, T indicates the time, where a value of 1 is assigned to the post-treatment period, and the coefficient of interest ($\beta_3$) is the interaction of treatment group assignment and the treatment period. We added several school characteristics to the analysis to account for observable characteristics that might account for differences in FAFSA completion rates. The complete model takes the form:

$$y_{it} = \beta_0 + \beta_1 X_i + \beta_2 T_t + \beta_3 X_i \times T_t + \beta_{4(class)} T_t + \beta_{5(FLRE)} T_t + \beta_{6(URM)} T_t + \beta_{7(susp)} T_t + \varepsilon_{it} \tag{2}$$

In Formula (2), the size of the graduating class, the proportion of free or reduced lunch eligibility, the percentage of underrepresented minority students enrolled, and the percentage of students suspended each year are included as controls. DID was used to

analyze the effects of several higher education policies [33], notably to evaluate the Georgia HOPE scholarship and the federal tax credit [34] and the adoption of state high school graduation requirement policies [35].

## 3. Results

The first results in Table 1 showed a trend comparison over the two months of the project, from the year before the intervention to the first year of implementation. During the two years under investigation here, the FAFSA application opened on 1 January; the figure shows that just a month into the filing season, 381 students already submitted their applications in 2012, and 422 submitted their applications during the year Say Yes announced the tuition guarantee and the FAFSA Completion project began. By February 15, a few of the applications submitted resulted from the project. The computer lab sessions in Phase I were already completed by then, but only students who qualified as independent could submit during those sessions. In both years, the scholarship fair in phase II was held by the end of January, and the fair accounted for nearly a third of submitted applications by 15 February.

**Table 1.** The difference in Differences Analysis Comparing Buffalo to Other Urban Centers, 2011–2013.

| Variable | B | SEB | Beta | Sig. |
|---|---|---|---|---|
| Cons. | 58.523 | 13.818 | 0.000 | *** |
| Year of FAFSA Completion | −5.526 | 4.608 | −0.062 | |
| Percent Free/Reduced Lunch | −0.661 | 0.213 | −0.213 | ** |
| PercentNon−White Students | −0.087 | 0.203 | −0.029 | |
| Percent Suspensions | −0.188 | 0.125 | −0.061 | |
| Number of HS Completers | 0.465 | 0.025 | 0.827 | *** |
| Treatment | −10.514 | 5.368 | −0.117 | ~ |
| DID | 30.151 | 6.816 | 0.283 | *** |
| F-test | 90.088 | | | *** |
| R2 | 0.910 | | | |

*** $p < 0.001$, ** $p < 0.01$, ~ $p < 0.10$.

Phase III began during the week of 22 February, when we expected more significant gains in FAFSA application submissions. For the next month, the rate of increase during the project exceeded the rate from 2012. During March, the rate of increased FAFSA submissions grew by 27% in 2012 and more than 35% in 2013. By the submission deadline for Say Yes to Education, the number of FAFSA applications submitted to the U.S. Department of Education grew nearly 62%. These data alone do not indicate whether the increase is attributable to the tuition guarantee, the FAFSA Completion project, or other factors. However, it does suggest a substantial increase that is worth exploring further.

There is another difference that we note descriptively, which is not pictured in the figure. FSA provided us with two numbers—FAFSA submitted and FAFSA Completed. The difference between the two numbers was that some applications were rejected because they either included errors or were missing signatures. If we take the ratio of completed applications to those submitted, we have an accuracy rate. In 2012, the federal government accepted approximately 87% of all applications submitted by BPS students. During the 2013 year, when the project was underway, those accuracy rates improved to 92%. We did not see accuracy rates improve across the other schools in this study. While we did not focus on this outcome, it may be most directly linked to the work of the FAFSA Completion project because individualized help was more likely to result in catching errors before the applications were submitted. In practical terms, that may amount to as many as 50 students in BPS schools whose applications were accepted that would not have been without the assistance of the project.

The first contrast compares outcomes for BPS students to those in Albany, Syracuse, or Rochester public schools. Table 1 summarizes the regression analysis, and the findings

can be understood relative to the total number of FAFSA forms submitted at a given school each year. Approximately 58 students submit the FAFSA each year across each high school included in the study. Schools with a higher enrollment of low-income students—as approximated by free or reduced lunch (FRL) eligibility—report a slightly lower average. A 10-percentage point decline in FRL would result in an average of five additional students completing the FAFSA. The difference-in-differences coefficient is an interaction term between year (pre- and post-program intervention) and treatment location (Buffalo vs. other three metro areas). The result compares BPS schools in the treatment year to BPS before the intervention and the three comparable mid-sized cities, which we expect to have a similar trend over time. The DID coefficient suggests that BPS schools saw an increase of 30 FAFSA applications submitted, compared to the trend for the other metro areas— an increase of approximately 50%, slightly less than the descriptive difference shown in Figure 1 above.

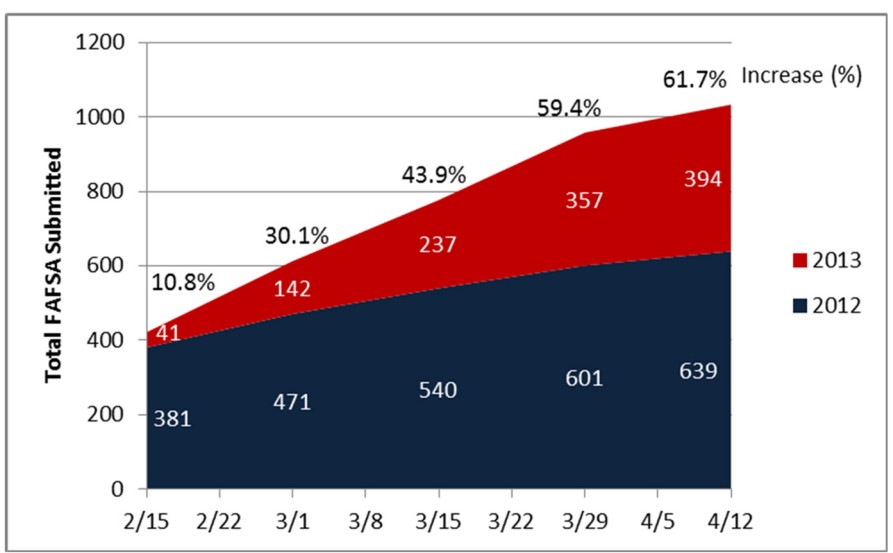

**Figure 1.** Comparing FAFSA Completion Rates in Buffalo, 2012–2013.

The second analysis in Table 2 compares BPS schools to the Buffalo public charter schools, which were eligible for the tuition guarantee but did not receive support from the FAFSA completion project during the first year. The sample size for the second analysis was much smaller because there are only sixteen BPS schools and five charter high schools in Buffalo, which accounts for the more modest *p*-values reported here. The mean number of applications submitted across all schools was 25, which was considerably lower than in the last analysis—and the reason is simple. The metropolitan areas outside of Buffalo have fewer high schools but are more extensive. A similar relationship exists between free and reduced lunch eligibility, indicating a 10-percentage point decrease in the free or reduced lunch eligibility would result in an additional eight students submitting their forms. The DID coefficient in this analysis suggests that BPS schools nearly doubled their FAFSA completion numbers from the first to the second year compared to the charter schools. The DID coefficient suggests that controlling for other school characteristics, an additional 24 students per school completed their FAFSA in Buffalo schools.

**Table 2.** The Difference in Differences Analysis Comparing BPS Schools to Buffalo Charter Schools, 2011–2013.

| Variable | B | SEB | Beta | Sig. |
|---|---|---|---|---|
| Cons. | 25.045 | 13.547 | 0.000 | ~ |
| Year of FAFSA Completion | 0.200 | 8.287 | 0.003 | |
| Percent Free/Reduced Lunch | −0.810 | 0.255 | −0.361 | ** |
| Percent Non−White Students | 0.338 | 0.251 | 0.171 | |
| Percent Suspensions | −0.399 | 0.188 | −0.160 | * |
| Number of HS Completers | 0.544 | 0.052 | 0.758 | *** |
| Treatment | −3.900 | 6.848 | −0.049 | |
| DID | 24.425 | 9.495 | 0.352 | * |
| F−test | 34.879 | | | *** |
| R2 | 0.878 | | | |

*** $p < 0.001$, ** $p < 0.01$, * $p < 0.05$, ~ $p < 0.10$.

The analyses reported above provide evidence to suggest two key outcomes. First, it appeared that the announcement of the tuition guarantee influenced students' FAFSA submission behavior, which should not be a surprise given that it is a condition of eligibility. When comparing BPS to other mid-sized cities in NY, Buffalo increased the number of students completing the FAFSA, even after controlling for key demographic characteristics of schools. In the descriptive analysis, we reported a change of 62% from the year before the announcement of the Say Yes tuition guarantee to the first year of implementation. The comparison with Syracuse, Rochester, and Albany indicated a 50% increase after controlling for other factors, which was similar in magnitude. The comparison with the public charter schools in Buffalo may indicate the independent effects of the FAFSA completion project because both groups were eligible for the tuition guarantee. However, only BPS received the FAFSA completion support.

It is important to recognize that the differences reported here were by April 12 of that year. We know that in the past, many students would have completed their FAFSA in August, right before the start of college—and many of them were attending community colleges. The advantage of setting the deadline for submission is that students are forced to complete the process sooner when help is available. They receive more information about higher education costs sooner, which makes the college choice process more manageable. What we did not change in any meaningful way were the factors that lead to many students making last-minute decisions about whether or where to attend. For example, some students were unclear whether they would graduate high school, and some required additional summer coursework. Our goal with the project was not to suggest that every student should go to college but to keep the option open to as many students as possible.

## 4. Discussion

The program's potential effects are an essential part of the experience in Buffalo. Our findings on the FAFSA completion project provide strong evidence that a personalized intervention can increase the likelihood students will complete a financial aid application, particularly in a high-need district like Buffalo. This has significant implications for other large metropolitan centers in the U.S.A. and potentially other countries with higher tuition prices for tertiary education [36]. The project was effective in the Buffalo context, because it was developed and implemented in the context of a larger collective impact strategy designed to leverage the power of the school district, local colleges and universities, and participating NGO's. However, the more important implications for researchers, community educators, and NGO's is the collaboration that led to this intervention. We believe there are several important lessons for other local communities, regardless of state or national context, about effective collaborations among schools, colleges, universities, and local community-based organizations.

*4.1. Invest Time to Develop Strong Collaborative Relationships*

Perhaps the most important lesson from this program is that collaboration takes time, and it is easiest to cultivate those relationships when the stakes are lower. The opportunities to establish trust are higher. Researchers from UB were already engaged in work around high school dropout prevention well before we began the pilot for the financial aid application process. At that time, dropout prevention was a priority for the district, and our participation helped develop a level of trust necessary when we spent six months shadowing school counselors. The time we spent shadowing counselors was instructive because we had an opportunity to identify problems of practice, all related to the administration of the college choice process. We saw firsthand how much time and energy counselors spent working with students on college applications, essays, admissions test registrations, college visits, fee waivers, and financial aid. We understood the challenges from their perspective and set out to develop solutions that were responsive to the needs of counselors and adapted to the work they were already doing to solve many of these problems. We focused on financial aid in the pilot study because this was part of the college choice process that made counselors most nervous. School counselors were less knowledgeable about financial aid and apprehensive about managing student and family personal data, including social security numbers and tax documents. No matter what problem we hope to address, in partnership with schools, researchers must spend time understanding the experiences of their collaborators and establishing trust.

*4.2. Translate Research into Practice*

As partners from a research university, our most important contribution was to be able to bring research to bear on the problems we were trying to address. As such, it was beneficial for us to be immersed in the research on college access and choice to identify research-informed strategies to address the complexity of the financial aid process. Perhaps the most notable contributions were the findings from Bettinger and colleagues that information-only strategies are not effective, at least in terms of improving college participation or financial aid application behavior, and the evaluative work of College Goal Sunday finding that the program was not reaching the students and families it was designed to serve. Financial aid information nights tended to be the most frequent intervention, mainly because they were inexpensive to provide. We could change our strategies to more active interventions because the research was compelling.

*4.3. Collaborate with Trusted Community-Based Partner Organizations*

Say Yes to Education played a critical role in the success of the collaboration between BPS and the university. As a backbone organization in a collective impact strategy, its primary responsibility is to manage the collective's shared agenda. They very effectively built trust among BPS leaders and local community partners. In many large districts, multiple partners provide similar services in schools with little coordination. Say Yes brought those partners together to identify ways to work together, and we found that the most effective mechanism was data sharing. Every pre-college program in Buffalo assisted its students and families with the financial aid application process. We found that sharing data across partner organizations incentivized more program providers to collaborate more effectively with school counselors.

*4.4. Listen to the Experiences and Needs of Education Partners*

Finally, and perhaps where we started, it was essential to listen to school counselors and tailor our strategies to the school's needs. We started by spending the first six months at a single school to better understand counselors' challenges in helping students with the college choice process. Then, we spent a semester learning with and from the school counselors at our pilot project site to understand better whether our design worked in the school context and how to adapt it to changing circumstances. When we began scaling the project up to serve the entire district rather than a single school, we conducted a site visit at

each school to discuss the plan with the school counseling teams. Each school followed different schedules, identified different classes that would be optimal for the computer lab portion of the intervention, and had different strategies for connecting students with the FAFSA completion volunteers. That time was necessary for the earliest stages of our work. While the trust established from the shadowing experiences and the advocacy of our partner counselors was helpful, we found it beneficial to continue to earn that trust with each set of school partners. Once the project concluded, we conducted follow-up visits with each school to learn what worked and could be improved. This formative evaluation was equally important for managing and sustaining relationships while refining the program to meet the needs of each school.

Effective collaboration takes time and, in our experience, cannot be rushed. We did not begin the relationship with a solution in search of a problem. Instead, we worked with the district to identify the problems most salient to them and coordinated with partners to develop, test, and refine a strategy to simplify the financial aid application process and increase the number of students eligible for the Say Yes tuition guarantee. Our work was possible because we constantly communicated with partners and used data to shape our work. The project is entering its tenth year, and the partnership continues to strengthen and grow. The project looks very different today than in the first year of implementation. Some of that is a consequence of the learning that occurred through the formative evaluation process. However, several program changes resulted from modifications to the financial aid application, including moving the opening of the application from January 1 to October 1, shifting from the PIN to the Federal Student Aid (FSA) ID username and password, and launching the IRS tax retrieval tool.

The collaboration among Say Yes, BPS, and the University at Buffalo established a community support network that attends to the six dimensions St. John identified in the human capabilities framework. Our focus in this study was on a specific and targeted intervention, but the larger collective impact initiative attends to the financial well-being of families. In partnership with the department of social services (DSS), Say Yes places a social worker in every school to identify and respond to the emergent needs of students and their families. The health clinics, legal clinics, and the growing mental health support are all related to the family's financial well-being because most would not have the resources to seek these services independently. During the pandemic, Say Yes led an effort to make laptops available to families that did not have the technology and to develop solutions for families that did not have high-speed internet access in their homes. Our work on the financial aid application process was not central to the preparation students received for college but Say Yes partnered closely with BPS to provide extended learning time after school, on weekends, and in the summer; they developed strategies to improve early childhood preparation; and they are assisting the district with some of their professional development needs. The entire program is predicated on improving college opportunities and Say Yes leveraged its relationships to establish a social network to support students through that process. They worked with employers to identify pathways through college or directly into the workforce. They engaged partners in the schools, community centers, churches, and the business community to make the promise of postsecondary education pay off for the students in the hope that they will return to the community and pay it forward to those that follow.

**Funding:** This research received no external funding.

**Institutional Review Board Statement:** Not applicable.

**Informed Consent Statement:** Not applicable.

**Data Availability Statement:** The data for this study are publicly available through Federal Student Aid and public archives can be found at https://studentaid.gov/data-center/student/application-volume/fafsa-completion-high-school (accessed on 1 May 2023).

**Conflicts of Interest:** The author declares no conflict of interest.

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
