# Peer review of "Say Yes to Education—Buffalo: A Human Capabilities Approach to College Access and Local Economic Development"

_education, doi:10.3390/educsci13050472_

Round 1
Reviewer 1 Report
Dear authors, I like your research topic but have some recommendations and questions.
1. Introduction section does not include related studies and should be added them.
2. Materials and Methods do not include the research model and should be added it.
3. How did you analyze the data?
4. What do you recommend for future implications and practices?
5. You should more international sources for the discussion section.
I have no recommendation.
Author Response
Thank you for the feedback on my paper. I have made the following changes to the paper based on your feedback.
I stated the research questions explicitly on page 2.
The reviewer asked that I include related studies in the first section and that may have been more a structural difference. I review the related research in the next two sections. In order to address their concern, I more clearly state where I am reviewing relevant research and then in those sections, I have made edits to more clearly indicate the literature I have reviewed and the relative strengths and limitations of the existing work. This is primarily an empirical paper looking at the effects of the intervention on financial aid applications so more focus is on that literature, but I also review some research related to university-community partnerships.
I added a brief section to discuss the implications of the FAFSA completion work itself in the discussion section. The majority of the discussion is focused more on what we learned as practitioner/scholars in the process of collaboration. I tie the discussion of the FAFSA completion project into the broader discussion of the university community collaboration that made the work possible and can generalize to many more communities and issues, irrespective of national context.
Finally, I added a reference to Bruce Johnstone who is widely regarded as one of the experts in international finance of postsecondary education and the complexities of cost-sharing among key constituents, including governments and individuals.
Thank you for the feedback. I believe it is a better paper as a result.
Reviewer 2 Report
This research is a relevant study. The theoretical foundation is adequately developed. Methodologically it is well structured, with a correct data analysis. The data analysis uses adequate statistics to respond to the object of the study. The conclusions respond to the purpose of the research.
Author Response
Thank you for your feedback.